# Automated Risk Prediction of Post-Stroke Adverse Mental Outcomes Using Deep Learning Methods and Sequential Data

**DOI:** 10.3390/bioengineering12050517

**Published:** 2025-05-14

**Authors:** Chien Wei Oei, Eddie Yin Kwee Ng, Matthew Hok Shan Ng, Yam Meng Chan, Vinithasree Subbhuraam, Lai Gwen Chan, U. Rajendra Acharya

**Affiliations:** 1Management Information Department, Office of Clinical Epidemiology, Analytics and kNowledge (OCEAN), Tan Tock Seng Hospital, Singapore 308433, Singapore; chienwei001@e.ntu.edu.sg; 2School of Mechanical and Aerospace Engineering, Nanyang Technological University, Singapore 639798, Singapore; 3Rehabilitation Research Institute of Singapore, Nanyang Technological University, Singapore 308232, Singapore; 4Department of General Surgery, Vascular Surgery Service, Tan Tock Seng Hospital, Singapore 308433, Singapore; 5The Digital Health Hub, Austin, TX 78944, USA; 6Department of Psychiatry, Tan Tock Seng Hospital, Singapore 308433, Singapore; 7Lee Kong Chian School of Medicine, Nanyang Technological University, Singapore 308232, Singapore; 8School of Mathematics, Physics and Computing, University of Southern Queensland, Brisbane, QLD 4305, Australia

**Keywords:** artificial intelligence, deep learning, machine learning, neural network, post-stroke anxiety, post-stroke depression

## Abstract

Depression and anxiety are common comorbidities of stroke. Research has shown that about 30% of stroke survivors develop depression and about 20% develop anxiety. Stroke survivors with such adverse mental outcomes are often attributed to poorer health outcomes, such as higher mortality rates. The objective of this study is to use deep learning (DL) methods to predict the risk of a stroke survivor experiencing post-stroke depression and/or post-stroke anxiety, which is collectively known as post-stroke adverse mental outcomes (PSAMO). This study studied 179 patients with stroke, who were further classified into PSAMO versus no PSAMO group based on the results of validated depression and anxiety questionnaires, which are the industry’s gold standard. This study collected demographic and sociological data, quality of life scores, stroke-related information, medical and medication history, and comorbidities. In addition, sequential data such as daily lab results taken seven consecutive days after admission are also collected. The combination of using DL algorithms, such as multi-layer perceptron (MLP) and long short-term memory (LSTM), which can process complex patterns in the data, and the inclusion of new data types, such as sequential data, helped to improve model performance. Accurate prediction of PSAMO helps clinicians make early intervention care plans and potentially reduce the incidence of PSAMO.

## 1. Introduction

### 1.1. Background

One of the main contributors to mortality and morbidity of the world population is stroke. In an article published by Global Stroke Factsheet in 2022, one in four people are estimated to have a stroke in a lifetime. The lifetime risk of developing stroke has increased by 50% over the last 17 years [1]. Based on reports from the Singapore Stroke Registry in 2020, the number of stroke episodes increased from 5890 in 2010 to 8846 in 2020, a significant rise in the past decade [2].

The prognosis of stroke varies widely and is influenced by several factors, including the type of stroke, severity, location of brain injury, patient age, pre-existing comorbidities, and the timeliness and quality of medical intervention.

Stroke can be classified into two types: ischemic and hemorrhagic. The prognosis of ischemic and hemorrhagic strokes differs primarily in the pattern of recovery, risks of complications, and rehabilitation needs. In ischemic stroke, prognosis is shaped by the extent of brain tissue damage caused by the lack of blood flow and how quickly circulation is restored. Hemorrhagic stroke prognosis, on the other hand, is influenced by the severity and location of bleeding, as well as the resulting pressure on brain structures. Recovery is often less predictable due to the potential for acute complications [3].

Pre-existing comorbidities, such as diabetes mellitus (DM), can significantly impact the prognosis of stroke by increasing the risk of complications. DM increases the risk of stroke by damaging blood vessels and promoting clot formation. In the context of stroke, diabetic patients often experience larger areas of brain injury and slower functional recovery. Their prognosis tends to be more complicated due to poor wound healing, increased risk of infections, and fluctuating blood sugar levels that can affect brain recovery. Additionally, they are more prone to recurrent strokes and long-term cognitive decline, making post-stroke rehabilitation more prolonged and challenging [4].

While some individuals experience significant recovery, especially with early rehabilitation, others may suffer from long-term impairments in mobility, speech, cognition, or emotional regulation [2].

Many stroke survivors are burdened by adverse outcomes such as depression and anxiety. Literature has termed it as post-stroke depression (PSD) and post-stroke anxiety (PSA), collectively known as post-stroke adverse mental outcomes (PSAMO). It is estimated that 20% of stroke survivors develop anxiety in their lifetime [5,6] and about 30% experience depression [5,7]. Stroke survivors with PSAMO experience negative health outcomes, such as higher mortality rate [8,9] and poorer functional disability [10,11]. The presence of untreated post-stroke anxiety or depression can place significant emotional and practical strain on family members and caregivers, potentially leading to burnout, reduced quality of care, and interpersonal conflict within the support system.

Studies have also shown that there are shared factors that contribute to the incidence of PSA and PSD, such as left hemisphere lesions and the presence of cognitive impairment [12]. Early treatment using antidepressants and psychotherapy in the early stages of stroke can be helpful to manage the onset of PSAMO [13,14,15]. Therefore, a prediction model is important to accurately predict the risk of PSAMO.

In addition, artificial intelligence (AI) has also gained momentum and has its fair share of success, specifically in the field of healthcare. Deep learning (DL) algorithms have proven to be successful and adopted to develop predictive models for disease prediction [16,17], especially in healthcare, where unstructured data types such as text in clinical notes [18] or images in x-rays [19] and other forms of scans are commonly used in the industry.

### 1.2. Literature Review

Previous studies have employed artificial intelligence (AI) to predict PSA, PSD, and PSAMO. Most studies used ML algorithms to develop the model. Wang et al. employed multi-layer perceptron (MLP) for the prediction of PSA [20]. However, the evaluation metrics of Wang et al. is the Euclidean distance between the predicted and actual anxiety scores of the patient, which differs from the objective of this study being a classification model. Oei et al. used ML algorithms to predict the risk of PSAMO and has yet to explore using deep loearning (DL) algorithms and methods [21,22,23].

In summary, most of the studies mentioned above are mainly ML models, and they only used static data for modelling [20,21,22,23]. Previous studies have yet to analyze using different data types and complex DL algorithms to develop a predictive model for PSAMO. A summary of the studies reviewed can be found below (Table 1), with each study highlighting the model that exhibits the best performance.

### 1.3. Motivation and Research Gap

This study has identified two research gaps. Firstly, DL methods have not been largely explored in this area, and this study will be the first to do so. AI has gained popularity by demonstrating its ability to model complex and challenging patterns. The development of AI models has been effective in applications to healthcare problems, such as prediction of readmission, etc. [24,25,26]. DL methods have also been applied specifically to detect and predict the onset of depression [27,28] and anxiety [29]. Thus, the first objective of this study is to leverage DL to develop a model capable of predicting the risk of PSAMO.

Secondly, most developed models use static data from a single time point. The study of stroke, depression, and anxiety is dynamic and longitudinal [30]. In our analysis of the literature, there was no model that could ingest and use other data types that complement static data for modelling.

Several studies have identified a significant association between elevated white blood cell (WBC) counts and symptoms of depression and anxiety. This relationship reflects the importance of the role of systemic inflammation in mood disorders [31,32]. Hence, the second objective of this study is to adapt and use different data types, such as longitudinal clinical variables such as daily lab results (e.g., white blood count, C-reactive protein, etc.), which are modeled using DL algorithms to capture temporal trends and interactions, hypothesising that including data from various data types leads to better model performance [33].

### 1.4. Main Contributions

To the best of our knowledge, our study is the first to use DL models to predict the risk of PSAMO instead of PSA and PSD separately. It is also the first to leverage the capabilities of DL to explore using different data types to develop a predictive model for prediction. The developed model is trained on a 179-subject dataset and has one of the highest accuracies and AUROC thus far. Accurate prediction of PSAMO allows clinicians to formulate care plans at the early stages of stroke and prevent the incidence of PSAMO.

## 2. Methods

### 2.1. Data Collection and Study Design

Patients who had been admitted for ischemic or hemorrhagic stroke to a tertiary care hospital are included in the study. The study period is from 2010–2021. These patients would have completed an anxiety or depression screening assessment, assessed within 7–37 days from the time of stroke. The 30-day time window of prediction is consistent with previous literature for identifying PSAMO during stroke recovery [34,35,36,37]. Patients who expired before 37 days were excluded from the study. Given the retrospective nature of this study, missing data is a common limitation. Therefore, to preserve data integrity, patients with an excessive proportion of missing values were also excluded from the analysis. In total, there were 179 patients, with 41 of them experiencing PSAMO and 128 without. The study was approved by the ethics review board. A summarized workflow can be found below (Figure 1).

Features extracted include baseline characteristics such as social information (e.g., educational level, occupation, etc.), quality of life scores, stroke-related information (e.g., type of stroke, the side where the stroke occurred, etc.), and medical and medication history. Baseline characteristics, collected at the patient’s admission date, are information that remains constant and does not change over a long period. They will be referred to as “static data” in this study.

In addition, this study also collected lab results for 7 consecutive days daily from the patient’s admission date, which will be referred to as “sequential data” in this study. These lab results are usually taken in the day daily to monitor the patient. The motivation to add lab results to the study partially comes from a separate study done by Qiu et al., who showed that biomarkers are a good predictor of depression [38] and, hence, may be useful predictors in our study as well.

Moreover, DL holds an advantage over ML methods, as it can model both static and sequential data. One of the objectives of this study is to use DL methods to capture signals from lab results over a specific time period, which may be valuable for making PSAMO predictions. A list of features that were used to train the model can be found in Appendix A.

### 2.2. Identification of PSAMO

The identification of PSA, PSD, or PSAMO was completed through the Hospital Anxiety and Depression Scale (HADS) and the Patient Health Questionnaire (PHQ). HADS uses a 14-item questionnaire that assesses anxiety and depression symptoms in medical patients. PSA and PSD were diagnosed if a patient scored above 7 on the anxiety scale and likewise on the depression scale (82% sensitivity and 78% specificity [39]). PSAMO is diagnosed if a patient has a combined score above 10.

PHQ has two versions: PHQ-9 and PHQ-2. PHQ-9 is a 9-item questionnaire assessing if a patient has depression symptoms; a score above 8 indicates PSD (88% sensitivity and 86% specificity) [40]. PHQ-2 is a simplified version of PHQ-9, containing only 2 questions; a score above 3 denotes PSD (83% sensitivity and 92% specificity) [33,41]. If one of the above criteria is met, the stroke patient is defined to have PSAMO; otherwise, no PSAMO.

### 2.3. Data Preprocessing and Engineering

Features with more than 25% of missing values were excluded from modelling. As the study is retrospective in nature, missing data is inevitable, mainly due to the loss of follow-up. Multiple imputation-chained equations were used to impute the remaining features [42]. The data is split into train and test sets in a ratio of 70:30. The training set employed a 10-fold cross-validation with 4 repeats to train and validate the model. Test set acted as unseen data and was used to evaluate model performance after training. Continuous features were standardized for the training set and applied to the test set. Categorical features were embedded using embedding layers. Instead of one-hot encoding, embedding helps to represent each individual category into a dense vector, thus reducing the dimensions of the dataset.

A total of 65 features were used to train the model; 46 are static data, while the remaining 19 are sequential data. The list of features can be found in Appendix A.

### 2.4. Model Developlemt

The model was developed using a multitude of DL architectures. Each DL architecture has its own uniqueness. DL architectures also allow the flexibility of combining data types for modelling. More details on each DL architecture are explained below. The model is trained on a laptop with a graphic processing unit (GPU) that is CUDA-enabled, running on NVIDIA GeForce RTX 4070. The model is trained on a batch size of 10.

#### 2.4.1. Categorical Embedding

Categorical embedding is a form of representation learning that maps high-dimensional categorical variables into a lower-level representation using a dense vector [43]. It works by training a neural network model on a prediction task. During the model training process, the neural network learns the optimal vector representation for each category. These representations capture relationships and similarities between categories based on the patterns in the training data.

#### 2.4.2. Multi-Layer Perceptron

Multi-layer perceptron (MLP) is a type of artificial neural network combined by multiple layers of interconnected nodes [44]. An MLP consists of three or more layers: an input layer, an output layer, and one or more hidden layers. Each layer consists of weights, biases, and non-linear functions that serve as activating nodes for the other layers. Weights and biases are trained using backpropagation, an example of supervised learning [45].

An iteration of a training loop starts with the forward pass as described below with a general formula:$$z_{k}^{l}=\sum_{j}w_{kj}^{l}a_{j}^{l-1}+b_{k}^{l}$$$$a_{k}^{l}=\sigma (z_{k}^{l})$$

l denotes the index of the layer, k denotes the index of the neuron in layer l, $w_{kj}^{l}$ refers to the weight from neuron j to k in layer l, $a_{j}^{l-1}$ denotes the activation of neuron j in the previous layer, and $b_{k}^{l}$ is the bias of neuron k in layer l. σ is the activation function.

Once the forward pass runs through the whole neural network, the loss can be calculated based on the predicted value and the actual value. The general formula for loss is shown below:$$Loss=L(\hat{y},y)$$

L denotes the loss function, $\hat{y}$ is the predicted value, and y is the actual value.

Backpropagation then takes place with the intent of minimizing the loss functions with respect to its weights and biases. The gradients are computed using the chain rule as shown below:$$\frac{\partial L}{\partial w_{kj}^{l}}=\frac{\partial L}{\partial \hat{y}}\cdot \frac{\partial \hat{y}}{\partial z_{k}^{l}}\cdot \frac{\partial z_{k}^{l}}{\partial w_{kj}^{l}}$$$$\frac{\partial L}{\partial b_{k}^{l}}=\frac{\partial L}{\partial \hat{y}}\cdot \frac{\partial \hat{y}}{\partial z_{k}^{l}}\cdot \frac{\partial z_{k}^{l}}{\partial b_{k}^{l}}$$

The weights and biases are then updated using a gradient descent algorithm with a hyperparameter learning rate (α), using the formula below:$$w_{kj}^{l}\leftarrow w_{kj}^{l}-\alpha \frac{\partial L}{\partial w_{kj}^{l}}$$$$b_{k}^{l}\leftarrow b_{k}^{l}-\alpha \frac{\partial L}{\partial b_{k}^{l}}$$

The process is repeated depending on the number of iterations (epochs) until the model converges.

#### 2.4.3. Long-Short Term Memory (LSTM)

LSTM is based on a recurrent neural network (RNN)-based architecture, possessing the ability for sequence analysis and retaining temporal signals [46]. The structure of an LSTM cell can be found in Figure 2 below.

LSTM possesses a cell state, where it stores and carries information across different time steps. The cell state is influenced by the forget gate, which controls (using the sigmoid activation function) what information from the cell state should be retained or discarded. The uniqueness of LSTM is its ability to retain past information with its structure [47]. The hidden state in an LSTM cell acts similarly to hidden layers in MLP, where “short-term” information from the previous calculation steps is stored.

The complexity of the LSTM cell having many “gates” enables it to selectively learn, remember, and forget information over long sequences, providing the ability to model and capture long-term dependencies in sequential data [48]. Among these, the forget gate is especially important, as it controls what past information should be discarded from the cell state. By filtering out irrelevant or outdated memory, it prevents information overload and helps the model stay focused on what matters most. This selective forgetting also stabilizes training by reducing the risk of vanishing gradients, making LSTMs more effective in learning patterns across time [49].

LSTM models are particularly well suited for sequential data due to their ability to handle variable-length sequences and capture long-term dependencies. Furthermore, their architecture is designed to retain relevant information over extended time intervals, enabling them to learn and model long-term relationships within the data. This capability is especially valuable in scenarios where earlier inputs significantly influence future outputs, thereby enhancing the model’s contextual understanding and predictive accuracy. Like MLP, LSTM also employs backpropagation for training its weights and biases [50].

#### 2.4.4. Model Architecture

The proposed PSAMO predictive model uses a combination of MLP and LSTM layers, as shown in Figure 3 below. Static and sequential data will be ingested into the model separately. Categorical variables from the static data are first separated and embedded through the embedding layers. An embedding layer works by mapping high-dimensional, sparse input into dense, low-dimensional vectors that capture meaningful relationships. The model learns these relationships during training, allowing similar inputs to match with similar vector representations. The embedded representations are then combined with the remaining variables found in the static data and parsed through the MLP layer, also known as “Multi-layer Perceptron (static data)” in Figure 3 below. “Multi-layer Perceptron (static data)” has three linear layers and a dropout layer with 20% dropout to prevent the model from overfitting [51].

Sequential data will be ingested into the model through the LSTM layer, represented by the yellow box in Figure 3 below. The strength of LSTM being able to capture temporal patterns is deployed in this context to capture temporal signals in the patient’s lab results 7 days leading to the prediction window. The LSTM layer has three hidden layers and outputs an eight-dimensional vector representing the signals captured from the daily lab results.

The proposed deep learning architecture leverages a hybrid design that effectively integrates both static and sequential data through a multi-component structure. The “Multi-layer Perceptron (static data)” module is dedicated to modelling static features, enabling the network to capture complex, non-temporal relationships and interactions. In parallel, the LSTM layer is employed to model sequential data, allowing the architecture to retain temporal dependencies and contextual patterns over time. These two representations are subsequently fused, enabling the model to learn higher-order interactions between static and temporal information. This is represented by “Multi-layer Perceptron (combined)” in Figure 3. This approach enhances the model’s capacity to learn from heterogeneous inputs, ultimately improving predictive performance and generalizability across tasks that involve both temporal dynamics and fixed attributes.

The output layer will be a sigmoid function, predicting the probability of a stroke patient experiencing PSAMO. The output layer will be a sigmoid function, predicting the probability of a stroke patient experiencing PSAMO.

#### 2.4.5. Model Initialization

The weights and biases of MLPs are initialized using “Kaiming initialization” [52], which accounts for the non-linearity of activation functions, such as rectified linear unit (ReLU), which are used in the model. Bias is initialized with 0. The weights are initialized with a normal distribution of$$w_{l}\sim N(0,\frac{2}{n_{l}})$$

$n_{l}$ refers to the number of input neurons at layer l.

The weights and biases of LSTMs are initialized using “Xavier initialization” or “Glorot initialization” [53]. Bias is initialized to be 0.001. The weights are initialized with a uniform distribution of$$w_{ij}\sim U(-\frac{\sqrt{6}}{\sqrt{S_{in}+S_{out}}},\frac{\sqrt{6}}{\sqrt{S_{in}+S_{out}}})$$

$S_{in}$ is the size of the previous layer, and $S_{out}$ is the size of the current layer.

Initializing the weights and biases of the neural network helps in faster convergence and avoids vanishing and exploding gradients. If the weights are too small, the gradients may vanish, making learning slow. On the other hand, if the weights are too large, the gradients may explode, leading to numerical instability. Proper initialization helps maintain a balance and prevents the model from reaching a local minima or maxima at the early stages of training [54,55].

#### 2.4.6. Model Training

The model is trained using mini-batch gradient descent (MBGD) [56], employing a learning rate of 0.001 and 2000 epochs. MBGD has been proven to have faster convergence [57] and efficient use of memory, as it processes only a small batch of training data at a given time and produces better generalization, which can help the model avoid local minima [58]. The formula for MBGD that updates the weights can be found below:$$w_{i+1}=w_{i}-\alpha \cdot \nabla_{w_{i}}L(x^{i:i+b},y^{i:i+b};w_{i})$$

$w_{i}$ represents the weights of the layers in the model in iteration i. α represents the learning rate, and b denotes the size of a single batch of data. The objective of MBGD is to minimize the loss function L(..) with respect to w.

The training pipeline also included an early stopping criterion of 50 epochs of minimal (0.001) or no improvement of the loss in the validation set to prevent overfitting.

As the model is a classification model, binary cross entropy loss is used as a loss function [59,60]. The formula for binary cross entropy loss can be found below:$$-\frac{1}{N}\sum_{i=1}^{N}y_{i}\cdot \log\left(p\left(y_{i}\right)\right)+\left(1-y_{i}\right)\cdot log(1-p\left(y_{i}\right))$$

N refers to total samples, $y_{i}$ represents the actual class of sample i, and $\log\left(p\left(y_{i}\right)\right)$ is the probability of being classified into that class.

Binary cross entropy loss penalizes incorrect labelling of the data class by a model if deviations in probability occur in classifying the labels. The objective is to minimize binary cross entropy loss through backpropagation, where low loss values equate to high accuracy.

### 2.5. Model Evaluation

The model was evaluated using standard classification metrics, accuracy, and F1 score. Area under receiver operating characteristics (AUROC) curve was also reported to measure the degree of discrimination between the two groups [61].

### 2.6. Packages Used

The preprocessing and modelling done in this study were implemented using Python 3 [62,63,64,65,66]. Each package used and their functions are listed in Appendix B.

## 3. Results

The classification results are shown in Table 2 below. The train set is reported with a confidence interval from the 10-fold cross-validation, while the test set is the performance of the model on the unseen data. It is evident that the proposed model in this study (MLP with LSTM) obtained the highest accuracy and AUROC of 0.792 and 0.789, respectively, which is better than baseline traditional ML algorithms. The F1 score has also improved to 0.353, improving classification accuracy for the minority class as well. The confusion matrix of the classification can be found in Table 3 below.

The difference between the proposed model in this study and the traditional ML models is (1) the utilization of DL architectures and (2) the addition of sequential data (daily lab results) for model training, yielding better performance.

## 4. Discussion

### 4.1. Model Performance

Table 2 above shows the improvement in model performance by using DL methods (MLP and LSTM) compared to the ML model (gradient-boosted trees) developed by Oei et al. Accuracy, AUROC, and F1 score have improved to 0.792, 0.789, and 0.353, respectively. Benchmarking against the models published to date, our model attained higher accuracy and AUROC (see Table 1).

Table 4 below provides a summary and compares the proposed method in this study versus other studies. The improvement in performance can be attributed to two reasons.

First is the utilization of DL algorithms for modelling. As shown in Table 4, most literature has utilized only classical ML methods, such as support vector machines, which may not effectively capture the complexity of the underlying patterns [67]. The nature of DL algorithms being able to effectively capture complex non-linear relationships may be one of the factors that contribute to the improvement of the prediction [68,69,70]. However, MLP itself does not inherently capture sequential dependencies, limiting its ability to model sequential or time-series data [71]. Therefore, there is a need for LSTM architectures to be incorporated to model the sequential lab results data that were collected, as proposed in this work.

Second, adding sequential lab results data, along with the LSTM architecture, improves model performance. Using a combination of different data types for prediction has proven successful in healthcare, as done by Zhang et al. [33]. Our study suggests potential signals found in the daily lab results that might contribute to the prediction of PSAMO, leading to the prediction window. The robustness of DL architectures enables different data types to be harmonized and modelled together, facilitating the integration of diverse information sources into a unified framework for more comprehensive and accurate insights.

### 4.2. Related Works

This study demonstrated a modest improvement in predictive performance, as reflected in both AUC and accuracy metrics, compared to existing work in the field (Table 1). For instance, Ryu et al. reported an AUC of 0.711, whereas the present study achieved an AUC of 0.791. Additionally, the current study employed a substantially larger sample size (*n* = 179 vs. *n* = 65), enhancing the generalizability of its findings. Notably, while Ryu et al. utilized traditional machine learning algorithms, this study leveraged deep learning approaches capable of modeling sequential data, an aspect that may contribute to improved predictive capability. It is important to note, however, that Ryu et al. work focused exclusively on PSD, whereas the present study addressed a broader spectrum of PSAMO. As such, a direct one-to-one comparison between the two studies is limited in scope [22].

In comparison to the study conducted by Oei et al. [21], the present study incorporates a more comprehensive set of features and a richer dataset. The inclusion of sequential laboratory results and the transition from traditional ML models to DL architectures may have contributed to the observed improvement in predictive performance. This is evidenced by an increase in AUC from 0.620 (Oei et al.) to 0.789 in the current study. Nevertheless, it is important to acknowledge that Oei et al. study employed a significantly larger sample size (*n* = 1790 vs. *n* = 179), which enhances the external validity and generalizability of their findings [21].

Recent advancements in the application of “dynamic displacement” data to machine learning models have demonstrated considerable promise in a range of domains. Several studies have highlighted the value of time-evolving physiological signals (e.g., biomarker fluctuations) in enhancing predictive accuracy [72,73,74]. In the context of healthcare, where baseline values and deviations from baseline are clinically meaningful, dynamic modeling approaches have proven effective in capturing subtle physiological changes that may precede critical adverse events [75,76]. By accounting for patterns such as sustained elevations, abrupt inflections, and rates of change in laboratory parameters like WBC count or CRP, machine learning models can potentially produce better results. LSTM networks, like the one used in this study, are well suited to capture these evolving dynamics, thereby facilitating earlier clinical intervention and more precise risk stratification.

Using this study as an example, an acute elevation in WBC count can be interpreted as a clinically meaningful deviation from baseline, representing a form of temporal displacement that the model is trained to detect and learn from. The LSTM model is designed to implicitly learn such non-linear and time-dependent patterns, enabling it to distinguish between transient fluctuations and sustained trends. Future extensions of this work may explore the explicit engineering of displacement-related features, including the rate of change, cumulative deviation from baseline, and other time-sensitive transformations. Incorporating these elements has the potential to enhance both the predictive performance.

### 4.3. Advantages and Limitations

PSAMO is a complex and multi-disciplinary condition; indicators and signals that predict the onset of it may come from various data types. This study was also the first to combine data types for modelling the prediction of PSAMO. The inclusion of sequential time series data in the model has improved model prediction. The developed model in this study performs better than most models in predicting PSAMO. Past studies have also suggested that regular screening for depression helps to prevent the incidence of PSAMO. Therefore, improved accuracy of such prediction can yield better triaging of stroke patients, leading to better allocation of healthcare resources and better management of such conditions upstream [77,78].

Having a manually human-administered PHQ and HADS test is also subject to human errors [79], translation errors [80], and biases [81]. Developing a predictive model helps mitigate these issues by introducing a systematic and automated approach and reducing reliance on subjective interpretation [82].

However, the nature of using complex DL architecture has made the model unexplainable. The model was also tested on a smaller sample size due to missing data, and further validations with different sites are necessary for external validity. Future expansion of this study can include data from other data types, such as images or text, for a more comprehensive approach, as stroke, depression, and anxiety are multi-disciplinary conditions.

Though the model has proven that adding sequential data improves model performance, it is operationally intensive for data or results to be taken daily. Clinicians on the ground should carry out careful considerations to weigh between operational needs and model performance. Additionally, data quality is essential for the successful implementation of this method. As the approach is sensitive to missing or inaccurate data, the validity of the results may be affected by inconsistencies or gaps within the dataset.

A small sample size poses significant limitations, particularly in the context of complex operations like stroke patient management and hospital workflows. The diversity of stroke cases, varying recovery trajectories, and individualized treatment protocols make it challenging to capture a representative dataset. Moreover, the dynamic nature of hospital operations, influenced by fluctuating patient loads and resource constraints, introduces variability that a small sample size might fail to reflect accurately. This limitation can result in models or analyses that restrict generalizability.

### 4.4. Potential Implementation

The implementation of a predictive model such as what was mentioned in this paper offers several significant clinical and operational benefits. Early identification of patients at high risk allows for timely psychological assessment and intervention, which can prevent the escalation of symptoms and improve overall mental health outcomes. From a healthcare systems perspective, it enables more efficient allocation of mental health resources and supports personalized treatment planning. Additionally, early intervention can alleviate the emotional and logistical burden on caregivers, contributing to a more sustainable and supportive recovery environment. Overall, such a model promotes a holistic approach to stroke recovery by integrating mental health into routine clinical care.

However, to ensure effective implementation, several components must be addressed. First, clinical validation and external testing of the model are essential to confirm its predictive accuracy and generalizability across diverse patient populations. Without robust validation, there is a risk of overfitting or misclassification, which could lead to inappropriate care decisions. Second, training and awareness among clinical staff must accompany implementation so that providers understand the model’s purpose, limitations, and how to use its outputs responsibly [83]. The objective of the model should allow clinicians to work in tandem with the model and not replace clinical decisions.

Additionally, ongoing monitoring and recalibration of the model are critical to maintain its performance over time [84]. As patient demographics drift, care practices, or mental health trends evolve, the model must be updated to reflect these changes. Creating feedback loops where clinicians and researchers can evaluate predictions against real outcomes will strengthen the model’s reliability and support continuous improvement. These steps are vital for transforming a predictive model from a theoretical tool into a practical, effective part of stroke care that contributes to better long-term recovery outcomes.

## 5. Conclusions

In conclusion, this study has shown that using DL architectures and the addition of sequential time series data has improved model performance in predicting PSAMO with 79.2% accuracy. The improvement of model performance may lead to better predictions of PSAMO, better triaging of healthcare resources, and potentially reducing the incidence of PSAMO. Early detection and intervention can mitigate the emotional, physical, and logistical demands placed on caregivers, thereby enhancing their well-being and sustaining their capacity to provide long-term support. This study has yet to explore using other data types (e.g., text or images) for prediction, which can potentially be studied in the future and may improve model performance.

## Figures and Tables

**Figure 1 bioengineering-12-00517-f001:**
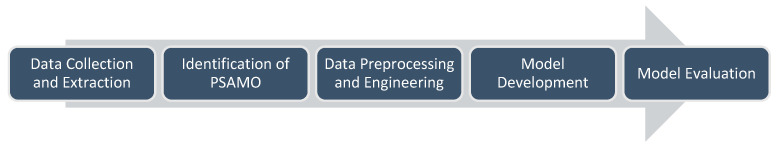
Flow diagram of the proposed model.

**Figure 2 bioengineering-12-00517-f002:**
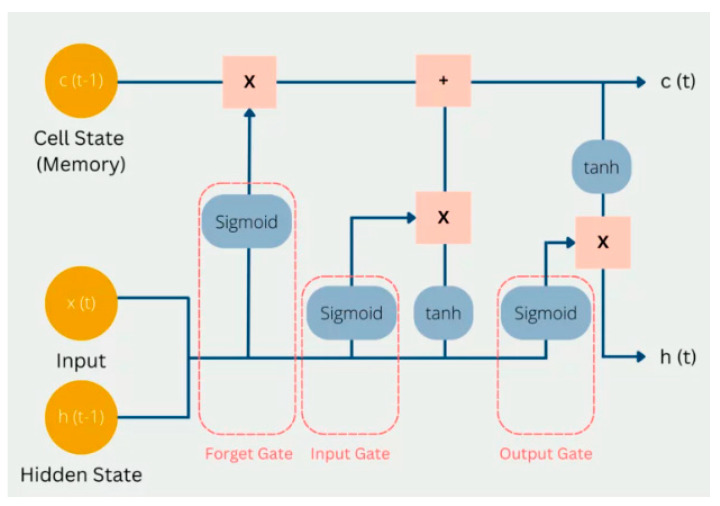
Illustration of an LSTM cell (extracted from: https://databasecamp.de/en/ml/lstms, accessed on 21 January 2025).

**Figure 3 bioengineering-12-00517-f003:**
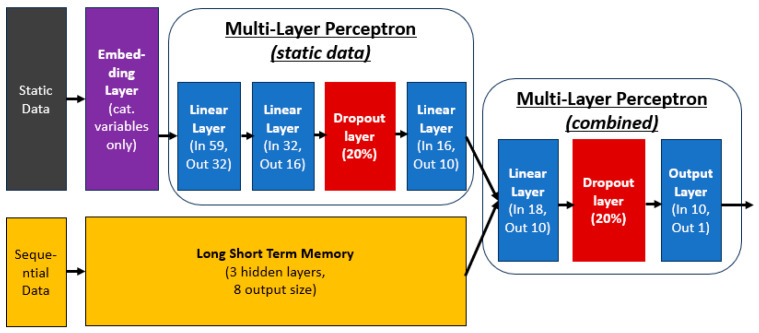
Model architecture—prediction of PSAMO.

**Table 1 bioengineering-12-00517-t001:** Summary of studies for automated prediction of post-stroke adverse mental outcome.

Author	Dataset	Features	Outcome	Techniques	Best Performance
Wang et al., 2021 [20]	395 cases	Demographics, lab results, vascular risk factors	PSA	RF, DT, SVM, stochastic gradient descent, multi-layer perceptron	Demographics, lab results, vascular risk factors
Oei at al., 2023 [21]	285 PSAMO and 1495 no PSAMO cases	Demographics, stroke-related data, surgical and medical history, etc.	PSAMO ^1^	Logistic regression, DT, GBT, RF, XGBoost, CatBoost, AdaBoost, LightGBM	GBT: AUC 0.620; Acc 0.747; F1 score 0.341
Ryu et al., 2022 [22]	31 PSD and 34 non-PSD cases	Medical history, demographics, neurological, cognitive, and functional test data	PSD	SVM, KNN, RF	SVM: AUC 0.711; Acc 0.70; Sens 0.742; Spec 0.517
Fast et al., 2023 [23]	49 PSD and 258 non-PSD cases	Demographics, clinical, serological, and MRI data	PSD ^1^	GBT, SVM	GBT: Balanced Acc 0.63; AUC 0.70

Acc, accuracy; AUC, area under the curve; DT, decision tree; GBT, gradient-boosted tree; KNN, k-nearest neighbor; RF, random forest; Sens, sensitivity; Spec, specificity; SVM, support vector machine. ^1^ Developed models were explainable.

**Table 2 bioengineering-12-00517-t002:** Model performance in prediction of PSAMO.

Developed Models	Train Set			Test Set		
	Accuracy	AUROC	F1 Score	Accuracy	AUROC	F1 Score
Gradient-Boosted Trees (Oei et al., 2023) [21] (Best Model using Classical ML Approach)	0.973 (0.958–0.982)	0.946 (0.932–0.957)	0.950 (0.924–0.964)	0.747	0.620	0.341
MLP + LSTM (Using both static and sequential data)	0.823 (0.721–0.852)	0.752 (0.621–0.784)	0.586 (0.328–0.622)	0.796	0.789	0.353

**Table 3 bioengineering-12-00517-t003:** Confusion matrix obtained from the test set.

		Predicted Label	
		Non-PSAMO	PSAMO
**Actual label**	**Non-PSAMO**	40 cases	0 cases
	**PSAMO**	11 cases	3 cases

**Table 4 bioengineering-12-00517-t004:** Summary of a comparison of methods for automated prediction of post-stroke adverse mental outcome.

Author	Data Type		Artificial Intelligence Methods		Outcome	Best Performance
	Static	Sequential	Machine Learning	Deep Learning		
Wang et al., 2021 [20]	√		√		PSA	18.625 Euclidean distance between anxiety scores
Oei at al., 2023 [21]	√		√		PSAMO ^1^	AUC 0.620; Acc 0.747; F1 score 0.341
Ryu et al., 2022 [22]	√		√		PSD	AUC 0.711; Acc 0.70; Sens 0.742; Spec 0.517
Fast et al., 2023 [23]	√		√		PSD ^1^	Balanced Acc 0.63; AUC 0.70
Current study	√	√		√	PSAMO	AUC 0.789; Acc 0.796; F1-score 0.353

Sens, sensitivity; Spec, specificity; SVM, support vector machine. ^1^ Developed models were explainable.

## Data Availability

Data is unavailable due to privacy restrictions.

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
