# Peer review of "Automated Risk Prediction of Post-Stroke Adverse Mental Outcomes Using Deep Learning Methods and Sequential Data"

_bioengineering, 2025, doi:10.3390/bioengineering12050517_

Round 1
Reviewer 1 Report
Comments and Suggestions for Authors
- In abstract, outlines objectives, methods, and outcomes. However, it would be beneficial to explicitly state the specific deep learning methods (e.g., MLP, LSTM) in the abstract itself.
- In introduction , introduces the research problem, highlighting the significance and prevalence of PSAMO.
- Table 1 is valuable, but clearly justify the choice of baseline comparison (Gradient Boosted Trees) among all other ML models previously reported.
- For data collection, please justify more explicitly the rationale for choosing a relatively small sample size (179 subjects). Discuss how representative this sample might be of broader stroke populations.
- In conclusion, summarizes the main findings and future work. Emphasize explicitly the practical impact this predictive model could have on clinical practice, particularly in early intervention strategies for stroke survivors.
Author Response
Dear Reviewer,
Thank you for spending time to review our manuscript. We have taken your comments into consideration and made the following edits. Please see our responses to the comments below:
In abstract, outlines objectives, methods, and outcomes. However, it would be beneficial to explicitly state the specific deep learning methods (e.g., MLP, LSTM) in the abstract itself.
- [Response] : Thank you for the comments. The deep learning methods have already been stated in the abstract section, under the results subsection. A snippet of the results subsection is pasted here for easy reference
“The combination of using DL algorithms, such as Multi-Layer Perceptron (MLP) and Long-Short Term Memory (LSTM), that can process complex patterns in the data and the inclusion of new data types, such as sequential data, helped to improve model performance. " [Pg 1, Line 32-34 of the manuscript]
In introduction , introduces the research problem, highlighting the significance and prevalence of PSAMO.
- [Response] : Thank you for the comments. We have added the significance of PSAMO and how it may potentially impact the social support system of the patient. The introduction has been edited to include the following:
“The presence of untreated post-stroke anxiety or depression can place significant emotional and practical strain on family members and caregivers, potentially leading to burnout, reduced quality of care, and interpersonal conflict within the support system.” [Pg 2, Line 59 - 61 of the manuscript]
Table 1 is valuable, but clearly justify the choice of baseline comparison (Gradient Boosted Trees) among all other ML models previously reported.
- [Response] : Thank you for the comments. The comparison was against the best performing algorithms that was experimented in each manuscript. Gradient Boosted Trees appears more frequently to be the better performing algorithm. We have added a section to justify that GBT is the best performing model in the manuscripts that we are comparing, for clarity. The Literature Review section has been edited to include the following:
“A summary of the studies reviewed can be found below (Table 1), with each study highlighting the model that exhibits the best performance.”[Pg 2, Line 85 - 86 of the manuscript]
For data collection, please justify more explicitly the rationale for choosing a relatively small sample size (179 subjects). Discuss how representative this sample might be of broader stroke populations.
- [Response] : Thank you for the comments. We have added in section 2.1 “Data Collection and Study Design” to justify that the relatively small sample size is partly because of missing data due to the retrospective nature of the study.
“Given the retrospective nature of this study, missing data is a common limitation. Therefore, to preserve data integrity, patients with an excessive proportion of missing values were also excluded from the analysis.” [Pg 4, Line 120 - 122 of the manuscript]
In conclusion, summarizes the main findings and future work. Emphasize explicitly the practical impact this predictive model could have on clinical practice, particularly in early intervention strategies for stroke survivors.
- [Response] : Thank you for the comments. Added in the conclusion section on how early detection and intervention can mitigate the emotional, physical, and logistical demands placed on caregivers, thereby enhancing their well-being and sustaining their capacity to provide long-term support . The conclusion section now includes:
“Early detection and intervention can mitigate the emotional, physical, and logistical demands placed on caregivers, thereby enhancing their well-being and sustaining their capacity to provide long-term support.” [Pg 12, Line 422 - 424 of the manuscript]
Reviewer 2 Report
Comments and Suggestions for Authors
SUMMARY
This article focuses on automated prediction of post-stroke adverse mental outcomes (PSAMO) using Deep Learning (DL) methods. The authors describe a predictive method to help physicians make early intervention plans and potentially reduce the incidence of PSAMO.
In general, the level of research and paper is high, but there are a number of comments.
COMMENTS
1. It is recommended to add a review of current research (2022-2024) on the use of deep neural networks for medical prediction.
2. It is unclear whether feature selection was performed manually or using automated methods (e.g., SHAP, PCA, Recursive Feature Elimination).
3. The key factors affecting prognosis need to be more clearly described.
4. Graphs (Figure 2, Figure 3) need additional explanation (e.g., why were certain features key?).
5. The article does not discuss how the findings can be implemented in clinical practice. The study is very relevant, but a discussion of implementation into practice would be worth adding.
6. It is also recommended to pay attention to the limitations of the method (e.g., dependence on data quality).
Author Response
Response to Reviewer
This article focuses on automated prediction of post-stroke adverse mental outcomes (PSAMO) using Deep Learning (DL) methods. The authors describe a predictive method to help physicians make early intervention plans and potentially reduce the incidence of PSAMO.
In general, the level of research and paper is high, but there are a number of comments.
- [Response]: Thanks for your comment and the time to read the manuscript. We are grateful that we are able to make a small impact in our research field. We have taken into considerations of the comments and have responded to them specifically below.
COMMENTS
1. It is recommended to add a review of current research (2022-2024) on the use of deep neural networks for medical prediction.
- [Response]: Thanks for your comment. We have added and commented on some lit reviews of the use of deep neural networks for medical prediction and added it in the background section. The background section in the manuscript now includes the following:
“In addition, Artificial Intelligence (AI) has also gained momentum and has its fair share of success, specifically in the field of health care. Deep Learning (DL) algorithms have proven to be successful and adopted to develop predictive models for disease prediction [14] , [15] especially in healthcare where unstructured data types such as text in clinical notes [16] or images in X-rays [17], and other forms of scans are commonly used in the industry.” [Pg 2, Line 67 - 72 of the manuscript]
- It is unclear whether feature selection was performed manually or using automated methods (e.g., SHAP, PCA, Recursive Feature Elimination).
- [Response]: Thanks for your comment. There is no feature selection performed. The only “selection” equivalent that we did was the dropping of features that have excessive missingness due to the retrospective nature of the study. For clarity, we have also added a small paragraph to explain the retrospective nature of the study:
“Given the retrospective nature of this study, missing data is a common limitation. Therefore, to preserve data integrity, patients with an excessive proportion of missing values were also excluded from the analysis.” [Pg 4, Line 120 - 122 of the manuscript]
- The key factors affecting prognosis need to be more clearly described.
- [Response]: Thanks for your comment. We have added a section on stroke prognosis in the introduction of the manuscript as well. The following section has been added:
“The prognosis of stroke varies widely and is influenced by several factors, including the type of stroke, severity, location of brain injury, patient age, pre-existing comorbidities, and the timeliness and quality of medical intervention. While some individuals experience significant recovery, especially with early rehabilitation, others may suffer from long-term impairments in mobility, speech, cognition, or emotional regulation [2].” [Pg 2, Line 48 - 52 of the manuscript]
- Graphs (Figure 2, Figure 3) need additional explanation (e.g., why were certain features key?).
- [Response]: Thanks for your comment. We agree with this and have added explanations to the 2.4.4 model architecture. The explanations also highlighted why did we choose certain architecture due to its key benefits and how they can attribute to better modelling of the prediction of PSAMO. The added section can be found here:
“LSTM models are particularly well-suited for sequential data due to their ability to handle variable-length sequences and capture long-term dependencies. Furthermore, their architecture is designed to retain relevant information over extended time intervals, enabling them to learn and model long-term relationships within the data. This capability is especially valuable in scenarios where earlier inputs significantly influence future outputs, thereby enhancing the model’s contextual understanding and predictive accuracy.” [Pg 6-7, Line 235 - 241 of the manuscript]
“The proposed deep learning architecture leverages a hybrid design that effectively integrates both static and sequential data through a multi-component structure. The “Multi-layer Perceptron (static data)” module is dedicated to modelling static features, enabling the network to capture complex, non-temporal relationships and interactions. In parallel, the LSTM layer is employed to model sequential data, allowing the architecture to retain temporal dependencies and contextual patterns over time. These two representations are subsequently fused, enabling the model to learn higher-order interactions between static and temporal information. This is represented by “Multi-layer Perceptron (combined)” in Figure 3. This approach enhances the model’s capacity to learn from heterogeneous inputs, ultimately improving predictive performance and generalizability across tasks that involve both temporal dynamics and fixed attributes. The output layer will be a sigmoid function, predicting the probability of a stroke patient experiencing PSAMO.” [Pg 7, Line 257 - 270 of the manuscript]
- The article does not discuss how the findings can be implemented in clinical practice. The study is very relevant, but a discussion of implementation into practice would be worth adding.
- [Response]: Thanks for your comment. We agree with this and have added section 4.3 Potential Implementation to the manuscript.
“4.3 Potential Implementation
The implementation of a predictive model such as what was mentioned in this paper offers several significant clinical and operational benefits. Early identification of patients at high risk allows for timely psychological assessment and intervention, which can prevent the escalation of symptoms and improve overall mental health outcomes. From a healthcare systems perspective, it enables more efficient allocation of mental health resources and supports personalized treatment planning. Additionally, early intervention can alleviate the emotional and logistical burden on caregivers, contributing to a more sustainable and supportive recovery environment. Overall, such a model promotes a holistic approach to stroke recovery by integrating mental health into routine clinical care.”
[Pg 11 to 12, Line 406 - 415 of the manuscript]
- It is also recommended to pay attention to the limitations of the method (e.g., dependence on data quality).
- [Response]: Thanks for your comment. We agree with this limitation and have added this comment to the limitation section of the manuscript. The snippet of the section that was added can be found here
“Additionally, data quality is essential for the successful implementation of this method. As the approach is sensitive to missing or inaccurate data, the validity of the results may be affected by inconsistencies or gaps within the dataset.” [Pg 11, Line 396 - 398 of the manuscript]
Reviewer 3 Report
Comments and Suggestions for Authors The subject is certainly very important. The methods and the presentation of the results are however suboptimal and need improvement. I would recommend a revision, where the authors address at least the points listed below. A clear plan of the paper must be presented in the end of the intro. After a very interesting and well-written intro, the transition to the models and to the interpretation of their results is rather unclear. The authors here need to clearly formulate the problem and justify why this particular approach is being used. It is also important to overview the state-of-the-art approaches and studies, that should be presented in detail. The model must be described in more details. The results are presented not clearly enough, sometimes even messy. Please focus on the presentation of your findings, the graphs must be prepared and described properly as well as the tables should be formatted optimally for presentation both in the PDF and in the online version of the paper. The authors are also encouraged to mention some recent machine-learning-based developments in the area of model prediction based on the analysis of displacement data. Here Refs. [https://www.science.org/doi/10.1126/science.abg1780] and [​​DOI: https://doi.org/10.1103/PhysRevResearch.5.043129] can be mentioned and the approaches used there in comparison to those of the current manuscript should be described and compared. I presume the authors want to examine certain causalities and inter-relationships between the stroke cases and the cases of subsequent depression. If so, this message needs to be communicated more clearly, also via reporting the related correlation coefficients. Finally, in the discussion the authors should return to the subject announced in the intro. What are the main results of the current study and how do they help solving the local as well as the global problems the authors posed at the start.Author Response
Response to reviewer 3
The subject is certainly very important. The methods and the presentation of the results are however suboptimal and need improvement. I would recommend a revision, where the authors address at least the points listed below.
- [Response]: Thanks for the comment and taking time to review our submission. We took these comments into considerations and will address them specifically below. The edits have already been completed in the manuscript, taking the below comments into account.
A clear plan of the paper must be presented in the end of the intro. After a very interesting and well-written intro, the transition to the models and to the interpretation of their results is rather unclear. The authors here need to clearly formulate the problem and justify why this particular approach is being used.
- [Response]: Thanks for the comment. We have added details to why we chose to leverage on AI to resolve this problem in the introduction section of the manuscript. We have also added details how the model can be part of a solution. Please see the following abstracts that are newly added to the manuscript:
- Specifically in the introduction, we added a section on the growing success of AI in the field of healthcare and specifically disease prediction:
“In addition, Artificial Intelligence (AI) has also gained momentum and has its fair share of success, specifically in the field of health care. Deep Learning (DL) algorithms have proven to be successful and adopted to develop predictive models for disease prediction [14] , [15] especially in healthcare where unstructured data types such as text in clinical notes [16] or images in X-rays [17], and other forms of scans are commonly used in the industry. “
[Pg 2, Line 67 - 72 of the manuscript]
This supports the motivation as specified in section 1.3 “motivation and research gap” of the manuscript. An extract of the manuscript can be found below:
“This study has identified two research gaps. Firstly, DL methods have not been largely explored in this area, and this study will be the first to do so. AI has gained popularity by demonstrating its ability to model complex and challenging patterns. The development of AI models has been effective in applications to healthcare problems, such as prediction of readmission, etc. [22], [23], [24]. DL methods have also been applied specifically to detect and predict the onset of depression [25], [26] and anxiety [27]. Thus, the first objective of this study is to leverage DL to develop a model capable of predicting the risk of PSAMO.” [Pg 2-3, Line 88 - 95 of the manuscript]
It is also important to overview the state-of-the-art approaches and studies, that should be presented in detail.
- [Response]: Thanks for the comment. We have added some literatures on how deep learning or artificial intelligence algorithms are used in the field of healthcare, specifically in disease prediction. They are appended into the manuscript in the introduction section. An extract of it can be found below:
“In addition, Artificial Intelligence (AI) has also gained momentum and has its fair share of success, specifically in the field of health care. Deep Learning (DL) algorithms have proven to be successful and adopted to develop predictive models for disease prediction [14] , [15] especially in healthcare where unstructured data types such as text in clinical notes [16] or images in X-rays [17], and other forms of scans are commonly used in the industry.”
[Pg 2, Line 67 - 72 of the manuscript]
The model must be described in more details.
- [Response]: Thanks for the comment. We have added details to the model architecture, its key features and the reason for using each specific feature in section 2.4.4 Model Architecture of the manuscript. The added section can be found here:
“The proposed deep learning architecture leverages a hybrid design that effectively integrates both static and sequential data through a multi-component structure. The “Multi-layer Perceptron (static data)” module is dedicated to modelling static features, enabling the network to capture complex, non-temporal relationships and interactions. In parallel, the LSTM layer is employed to model sequential data, allowing the architecture to retain temporal dependencies and contextual patterns over time. These two representations are subsequently fused, enabling the model to learn higher-order interactions between static and temporal information. This is represented by “Multi-layer Perceptron (combined)” in Figure 3. This approach enhances the model’s capacity to learn from heterogeneous inputs, ultimately improving predictive performance and generalizability across tasks that involve both temporal dynamics and fixed attributes. The output layer will be a sigmoid function, predicting the probability of a stroke patient experiencing PSAMO.” [Pg 7, Line 257 - 270 of the manuscript]
The results are presented not clearly enough, sometimes even messy. Please focus on the presentation of your findings, the graphs must be prepared and described properly as well as the tables should be formatted optimally for presentation both in the PDF and in the online version of the paper.
- [Response]: Thanks for the comment. We apologize for the inconsistency in the presentation of the results. We have taken note of the formatting issues in this revision.
The authors are also encouraged to mention some recent machine-learning-based developments in the area of model prediction based on the analysis of displacement data. Here Refs. [https://www.science.org/doi/10.1126/science.abg1780] and [​​DOI: https://doi.org/10.1103/PhysRevResearch.5.043129] can be mentioned and the approaches used there in comparison to those of the current manuscript should be described and compared.
- [Response]: Thanks for the comment. We have added some literatures on how deep learning or artificial intelligence algorithms are used in the field of healthcare, specifically in disease prediction. They are appended into the manuscript. This is also addressed above. For easy reference a snipper of the manuscript can be found here:
“In addition, Artificial Intelligence (AI) has also gained momentum and has its fair share of success, specifically in the field of health care. Deep Learning (DL) algorithms have proven to be successful and adopted to develop predictive models for disease prediction [14] , [15] especially in healthcare where unstructured data types such as text in clinical notes [16] or images in X-rays [17], and other forms of scans are commonly used in the industry." [Pg 2, Line 67 - 72 of the manuscript]
I presume the authors want to examine certain causalities and inter-relationships between the stroke cases and the cases of subsequent depression. If so, this message needs to be communicated more clearly, also via reporting the related correlation coefficients.
- [Response]: Thanks for the comment. The objective of this manuscript is more of leveraging on AI to develop a predictive model that can predict PSAMO. Therefore, we did not look at the related correlation coefficients as having high correlation coefficients may not attribute to better model performance.
Finally, in the discussion the authors should return to the subject announced in the intro. What are the main results of the current study and how do they help solving the local as well as the global problems the authors posed at the start.
- [Response]: Thanks for the comment. We have reviewed and edited the conclusion of the manuscript. We have mentioned that the model can potentially help to serve as an early prevention tool for PSAMO and in return provide better health outcomes for the stroke patients. An extract of the edited conclusion can be found here:
“Early detection and intervention can mitigate the emotional, physical, and logistical demands placed on caregivers, thereby enhancing their well-being and sustaining their capacity to provide long-term support.” [Pg 12, Line 422-424 of the manuscript]
Round 2
Reviewer 2 Report
Comments and Suggestions for Authors
SUMMARY
The authors provided an explanation of all comments and made adjustments to the manuscript, but there are still a number of comments.
COMMENTS
1. According to the previous comment "3. The key factors affecting prognosis need to be more clearly described", not only should a description of the factors (lines 48-52) be included, but also how they affect the prognosis and how they are accounted for.
2. According to the previous comment "4. Graphs (Figure 2, Figure 3) need additional explanation (e.g., why were certain features key?)", descriptions of Figures 2 and 3 have been added. But it is also recommended to explain why certain features were key. It is recommended to add how the model shown in Figure 3 works.
3. Section “4.3 Potential Implementation” is also recommended to be expanded. For example, add requirements for the implementation of the algorithm, not just the general theory.
Author Response
Dear Reviewer,
Thank you for the comments and also taking the time once again to review our manuscript. We have noted your comments and provided our responses below. For clarity, the edits to the manuscript this time will be highlighted in green instead of yellow previously.
COMMENTS
1. According to the previous comment "3. The key factors affecting prognosis need to be more clearly described", not only should a description of the factors (lines 48-52) be included, but also how they affect the prognosis and how they are accounted for.
Response: Thank you for the comments. We have expanded the section on the prognosis of stroke. Some key factors of prognosis such as the type of stroke (Ischemic and Hemorrhagic) and the comorbidities have been expanded and each provided with a paragraph of description.
Stroke can be classified into two types, Ischemic and Hemorrhagic. The prognosis of ischemic and hemorrhagic strokes differs primarily in the pattern of recovery, risks of complications, and rehabilitation needs. In ischemic stroke, prognosis is shaped by the extent of brain tissue damage caused by the lack of blood flow and how quickly circulation is restored. Hemorrhagic stroke prognosis, on the other hand, is influenced by the severity and location of bleeding, as well as the resulting pressure on brain structures. Recovery is often less predictable due to the potential for acute complications [3].
Pre-existing comorbidities such as Diabetes mellitus (DM) can significantly impact the prognosis of stroke by increasing the risk of complications. DM increases the risk of stroke by damaging blood vessels and promoting clot formation. In the context of stroke, diabetic patients often experience larger areas of brain injury and slower functional recovery. Their prognosis tends to be more complicated due to poor wound healing, increased risk of infections, and fluctuating blood sugar levels that can affect brain recovery. Additionally, they are more prone to recurrent strokes and long-term cognitive
The appended can be found in page 2, section 1.1, line 51 – 65, highlighted green in the manuscript.
2. According to the previous comment "4. Graphs (Figure 2, Figure 3) need additional explanation (e.g., why were certain features key?)", descriptions of Figures 2 and 3 have been added. But it is also recommended to explain why certain features were key. It is recommended to add how the model shown in Figure 3 works.
Response: Thank you for the comments. We have further expanded the explanation for Figure 2, highlighting the importance of the “Forget gate” and how the forget gate helps to stabilize training and filter out irreverent memory, which makes LSTM more effective.
Among these, the forget gate is especially important as it controls what past information should be discarded from the cell state. By filtering out irrelevant or outdated memory, it prevents information overload and helps the model stay focused on what matters most. This selective forgetting also stabilizes training by reducing the risk of vanishing gradients, making LSTMs more effective in learning patterns across time [49].
The appended section can be found in page 7, section 2.4.3, line 251 – 256, highlighted green in the manuscript.
Likewise, we have added additional explanations for Figure 3 as well to explain key features like the Embedding Layer and how the embedding layer works and capture meaningful representations of the data, making our overall architecture more effective.
An embedding layer works by mapping high-dimensional, sparse input into dense, low-dimensional vectors that capture meaningful relationships. The model learns these relationships during training allowing similar inputs to match with similar vector representations. The embedded representations are then combined with the remaining variables found in the static data and parsed through the MLP Layer, also known as “Multi-layer Perceptron (static data)” in Figure 3 below. “Multi-layer Perceptron (static data)” has 3 Linear Layers and a dropout layer with 20% dropout, to prevent the model from overfitting [51].
The appended section can be found in in page 7, section 2.4.4, line 268 – 275, highlighted green in the manuscript.
3. Section “4.3 Potential Implementation” is also recommended to be expanded. For example, add requirements for the implementation of the algorithm, not just the general theory.
Response: Thank you for the comments. We have added some key requirements for potential implementation of the model. Such examples include clinical validation of the model, training and awareness among clinical staff and also monitoring of models to prevent data draft.
However, to ensure effective implementation, several components must be addressed. First, clinical validation and external testing of the model are essential to confirm its predictive accuracy and generalizability across diverse patient populations. Without robust validation, there is a risk of overfitting or misclassification that could lead to inappropriate care decisions. Second, training and awareness among clinical staff must accompany implementation so that providers understand the model’s purpose, limitations, and how to use its outputs responsibly [83]. The objective of the model should allow clinicians to work in tandem with the model and not replace clinical decisions.
Additionally, ongoing monitoring and recalibration of the model are critical to maintain its performance over time [84]. As patient demographics drift, care practices, or mental health trends evolve, the model must be updated to reflect these changes. Creating feedback loops where clinicians and researchers can evaluate predictions against real outcomes will strengthen the model’s reliability and support continuous improvement. These steps are vital for transforming a predictive model from a theoretical tool into a practical, effective part of stroke care that contributes to better long-term recovery outcomes.
The appended section can be found in in page 12, section 4.3, line 442 – 456, highlighted green in the manuscript.

Reviewer 3 Report
Comments and Suggestions for Authors
The revision is not really substantial, the authors should try once more. The additions to the main text are not really extensive and essential. The added literature sources should also have been marked in the biblio. The proposed literature source on the interpretation of dynamical displacement data is not cited. A somewhat deeper revision, and particularly deeper discussion with a solid comparison with the state of the art, is necessary.
Author Response
Dear Reviewer,
Thank you for the comments and also taking the time once again to review our manuscript. We have noted your comments and provided our responses below. For clarity, the edits to the manuscript this time will be highlighted in green instead of yellow previously.
The revision is not really substantial; the authors should try once more. The additions to the main text are not really extensive and essential. The added literature sources should also have been marked in the biblio. The proposed literature source on the interpretation of dynamical displacement data is not cited. A somewhat deeper revision, and particularly deeper discussion with a solid comparison with the state of the art, is necessary.
We would like to address your comments in two main points.
Firstly, on “dynamical displacement” data. We would like to clarify that our study does not involve dynamical displacement data in the mechanical or structural sense. Instead, our work focuses on longitudinal clinical data, specifically daily laboratory results (e.g., white blood cell count, C-reactive protein, etc.), to model patient trajectories using a Long Short-Term Memory (LSTM) framework.
We have also taken into account the references that was suggested and also recognize how the study of dynamical displacement data can potentially be a predictive factor for PSAMO. Future work could explore the explicit modelling of these displacement-like dynamics by integrating derivatives (e.g., rate of change), cumulative deviation, or “return to baseline” metrics for relevant and significant biomarkers as features or interpretive layers, potentially enhancing explainability and clinical relevance. we have added a section on this in the discussion.
To address this, we have done the following:
- We have edited in the introduction section to explain in greater detail on the motivation behind using longitudinal clinical data. We hope that this will bring more clarity to the data being used in this study.
Several studies have identified a significant association between elevated white blood cell (WBC) counts and symptoms of depression and anxiety. This relationship reflects the importance of the role of systemic inflammation in mood disorders [31], [32]. Hence, the second objective of this study is to adapt and use different data types, such as longitudinal clinical variables such as daily lab results (e.g., White Blood Count, C-reactive protein, etc.), which are modeled using DL algorithms to capture temporal trends and interactions, hypothesizing that including data from various data types leads to better model performance [33].
This can be found in page 3, section 1.3, line 116 – 123 of the manuscript.
- We have also added section 4.2 Related works, where we have discussed on the successes of using dynamic displacement data and how we can potentially develop a model with such displacement-related variables. The section also discusses on how deviation from the baseline can be potentially useful to predict adverse critical events.
Recent advancements in the application of “dynamic displacement” data to machine learning models have demonstrated considerable promise in a range of domains. Several studies have highlighted the value of time-evolving physiological signal (such as biomarker fluctuations) in enhancing predictive accuracy [73], [74]. In the context of healthcare, where baseline values and deviations from baseline are clinically meaningful, dynamic modeling approaches have proven effective in capturing subtle physiological changes that may precede critical adverse events [75], [76]. By accounting for patterns such as sustained elevations, abrupt inflections, and rates of change in laboratory parameters like WBC count or CRP, machine learning models can potentially produce better results. LSTM networks, like the one used in this study, are well-suited to capturing these evolving dynamics, thereby facilitating earlier clinical intervention and more precise risk stratification. - Using this study as an example, an acute elevation in WBC count can be interpreted as a clinically meaningful deviation from baseline, representing a form of temporal displacement that the model is trained to detect and learn from. The LSTM model is designed to implicitly learn such nonlinear and time-dependent patterns, enabling it to distinguish between transient fluctuations and sustained trends. Future extensions of this work may explore the explicit engineering of displacement-related features, including the rate of change, cumulative deviation from baseline, and other time-sensitive transformations. Incorporating these elements has the potential to enhance both the predictive performance.
This can be found in page 12 section 4.2, line 427 – 446 of the manuscript.
Secondly, we have compared our models with several state-of-the-art approaches in the field, as presented in Table 1. A more in-depth discussion has been included in Section 4.2 (Related Works).
To address this point, we have taken the following actions:
- We have added Section 4.2, where we critically discuss and compare our study against several related works identified during our literature review. This section highlights the key differences in methodology, dataset composition, and modelling approach. We also provide an analysis of the factors that may have contributed to the observed improvements or limitations in our model performance relative to these studies. A section of what was added is shown below
This study demonstrated a modest improvement in predictive performance, as reflected in both AUC and accuracy metrics, compared to existing work in the field (Table 1). For instance, Ryu et al. reported an AUC of 0.711, whereas the present study achieved an AUC of 0.791. Additionally, the current study employed a substantially larger sample size (n = 179 vs. n = 65), enhancing the generalizability of its findings. Notably, while Ryu et al. utilized traditional machine learning algorithms, this study leveraged deep learning approaches capable of modelling sequential data, an aspect that may contribute to improved predictive capability. It is important to note, however, that Ryu et al.'s work focused exclusively on PSD, whereas the present study addressed a broader spectrum of PSAMO. As such, a direct one-to-one comparison between the two studies is limited in scope [20]. - In comparison to the study conducted by Oei et al., the present study incorporates a more comprehensive set of features and a richer dataset. The inclusion of sequential laboratory results and the transition from traditional ML models to DL architectures may have contributed to the observed improvement in predictive performance. This is evidenced by an increase in AUC from 0.620 (Oei et al.) to 0.789 in the current study. Nevertheless, it is important to acknowledge that Oei et al.'s study employed a significantly larger sample size (n = 1790 vs. n = 179), which enhances the external validity and generalizability of their findings [19].
This can also be found in Page 12, section 4.2 related works, line 408 – 446 of the manuscript.

Round 3
Reviewer 3 Report
Comments and Suggestions for Authors
Significant additional modifications: the material can be accepted in the present form.